# Identification, Characterization, and Stress Responsiveness of Glucose-6-phosphate Dehydrogenase Genes in Highland Barley

**DOI:** 10.3390/plants9121800

**Published:** 2020-12-18

**Authors:** Ruijun Feng, Xiaomin Wang, Li He, Shengwang Wang, Junjie Li, Jie Jin, Yurong Bi

**Affiliations:** Ministry of Education Key Laboratory of Cell Activities and Stress Adaptations, School of Life Sciences, Lanzhou University, Lanzhou 730000, China; fengrj16@lzu.edu.cn (R.F.); wangxiaomin@lzu.edu.cn (X.W.); hel16@lzu.edu.cn (L.H.); wangshw16@lzu.edu.cn (S.W.); jjli2017@lzu.edu.cn (J.L.); jinj2015@lzu.edu.cn (J.J.)

**Keywords:** abiotic stresses, glucose-6-phosphate dehydrogenase, highland barley, reactive oxygen species, redox homeostasis

## Abstract

G6PDH provides intermediate metabolites and reducing power (nicotinamide adenine dinucleotide phosphate, NADPH) for plant metabolism, and plays a pivotal role in the cellular redox homeostasis. In this study, we cloned five *G6PDH* genes (*HvG6PDH1* to *HvG6PDH5*) from highland barley and characterized their encoded proteins. Functional analysis of *HvG6PDHs* in *E. coli* showed that *HvG6PDH1* to *HvG6PDH5* encode the functional G6PDH proteins. Subcellular localization and phylogenetic analysis indicated that HvG6PDH2 and HvG6PDH5 are localized in the cytoplasm, while HvG6PDH1, HvG6PDH3, and HvG6PDH4 are plastidic isoforms. Analysis of enzymatic activities and gene expression showed that *HvG6PDH1* to *HvG6PDH4* are involved in responses to salt and drought stresses. The cytosolic *HvG6PDH2* is the major isoform against oxidative stress. *HvG6PDH5* may be a house-keeping gene. In addition, *HvG6PDH1* to *HvG6PDH4* and their encoded enzymes responded to jasmonic acid (JA) and abscisic acid (ABA) treatments, implying that JA and ABA are probably critical regulators of *HvG6PDH*s (except for *HvG6PDH5*). Reactive oxygen species analysis showed that inhibition of cytosolic and plastidic G6PDH activities leads to increased H_2_O_2_ and O_2_^−^ contents in highland barley under salt and drought stresses. These results suggest that G6PDH can maintain cellular redox homeostasis and that cytosolic HvG6PDH2 is an irreplaceable isoform against oxidative stress in highland barley.

## 1. Introduction

Drought and salinity are major abiotic stresses limiting plant growth and development, thus causing a significant reduction in crop yield [1,2]. Drought and salinity in plants impose severe water deficit, which causes the inhibition of essential enzymes, the decrease in photosynthesis, leaf transpiration rates, nutrient supply, as well as reactive oxygen species (ROS) accumulation [3,4,5]. Although ROS, as signaling molecules, can regulate diverse physiological and molecular processes [6,7,8], excess ROS can cause oxidative damage and affect the level and function of plant hormones, ultimately leading to cell death and protein modifications [9,10,11]. Therefore, the fine-tuned regulation of ROS homeostasis is the critical protective strategy against salt and drought stresses.

The oxidative pentose phosphate pathway (OPPP) is ubiquitous in eukaryotes. OPPP provides abundant intermediate metabolites and nicotinamide adenine dinucleotide phosphate (NADPH), which are necessary for the synthesis of nucleotide, amino acid, and fatty acid, carbon fixation as well as nitrogen assimilation [2,12,13,14,15,16,17,18]. Moreover, biosynthesis of the cellular redox buffer, reduced glutathione, depends on NADPH via glutathione reductase [6,19]. In plants, OPPP-associated NADPH synthesis is present in both cytoplasm and plastid through the rate-limiting glucose-6-phosphate dehydrogenase (G6PDH) [20,21]. Numerous literature have reported that G6PDH is involved in growth and development, such as oil and lipid accumulation, seed germination, and RNA biosynthesis [17,21,22]. Moreover, G6PDH responses to abiotic stresses such as drought [23,24], salt [25,26,27,28,29], metal [30,31], heat [32], cold [33,34], and so on; and in the dark, plastid G6PDH could provide NADPH instead of light reactions in the photosynthetic tissues [35,36]. Recently, studies of G6PDH mainly focus on its activity and gene expression, as well as the regulatory mechanism under various stresses, emphasizing the role of G6PDH in maintaining cellular redox homeostasis to enhance the tolerance to environmental stresses in plants.

Highland barley, also named “Qingke” in Chinese, is an important food crop and livestock feed and occupies 70% of crop lands in the Tibetan Plateau [37,38]. Compared to barley, which has covered caryopsis, highland barley has naked caryopsis and can adapt to extreme environmental conditions in Tibet with altitudes ranging from 2700 to 4000 m [37,38,39]. The majority of highland barley-growing areas expose to several environmental stresses, such as drought, low temperature, enhanced ultraviolet radiation, and salinity [39,40,41,42]. Our previous study has confirmed that G6PDH is a key regulator in the cross adaptation to salt and UV-B stresses by maintaining NADPH production in highland barley [40]. Moreover, G6PDH can regulate cellular redox balance and the response to oxidative stress by affecting the synthesis of antioxidant substances and activities of the plasma membrane (PM) H^+^-ATPase and Na^+^/H^+^ antiporter (SOS1) under salt and drought conditions [23,27,43,44]. However, the regulatory mechanism of the *G6PDH* gene family remains elusive in highland barley.

In the present study, we characterized five *G6PDH* genes in highland barley and analyzed their expression patterns and enzymatic activities. We found that *HvG6PDH1* to *HvG6PDH4* participate in response to salt and drought stresses and that the cytosolic HvG6PDH2 is the major isoform regulating cellular redox homeostasis. Moreover, jasmonic acid (JA) and abscisic acid (ABA) can positively regulate the expression of *HvG6PDHs* (except *HvG6PDH5*). Our data not only expand the information of the *G6PDH* gene family in plants but also offer new insights for metabolic adaptive mechanisms to abiotic stresses in highland barley.

## 2. Materials and Methods

### 2.1. Plant Material and Treatments

Seeds of highland barley (*Hordeum vulgare* L. *var. nudum Hook.f.*) cv Kunlun-14 were kindly provided by Qinghai Academy of Agricultural Sciences, Qinghai Province, China. The seeds of concordant size were surface sterilized with 2% NaClO (*v*/*v*), and then germinated overnight at 25 °C. The germinated seeds were carefully sowed into plastic containers filled with 1/4-strength Hoagland media [40]. All seedlings were grown in the greenhouse, where the photoperiod was 16 h light /8 h dark, and the temperature was set at 25 ± 2 °C with the relative humidity of 60–70% and 100 µmol·m^−2^·s^−1^ photon flux density.

Five-day-old seedlings were treated with the Hoagland medium supplemented with 150 mM NaCl (salt stress) or 20% PEG6000 (drought stress) for different times (3, 6, 12, 24, 48, 72, 96, 120, 144 h). For investigating the role of G6PDH on plasma membrane damage and the accumulation of active oxygen radicals (ROS), seedlings were first pretreated for 24 h with the Hoagland medium containing 10 mM glucosamine (Glucm), a competitive G6PDH inhibitor; and then, the seedlings were exposed to the Hoagland medium containing 10 mM Glucm (Glucm treatment), 150 mM NaCl and 10 mM Glucm (NaCl + Glum treatment), or 20% PEG 6000 and 10 mM Glum (PEG + Glum treatment) for 48 h, respectively. To examine the expression level of *HvG6PDHs* under phytohormone treatments, five-day-old seedlings were treated with ABA (100 µM) or JA (100 µM) for different times (3, 6, 12, 24, 48, 72, 96, 120, 144 h). The treatment solutions were replaced every two days. In the experiment, all the samples were collected at 9–10 o’clock in the morning and used immediately.

### 2.2. Identification of HvG6PDH Genes and Bioinformatics

The G6PDH sequences of Arabidopsis, maize, rice, and barley were searched in public databases, including HarvEST (http://harvest.ucr.edu/), Ensembl plant (http://plants.ensembl.org/Hordeum_vulgare/Info/Index), NCBI (www.ncbi.nlm.nih.gov), and BarleyGene Index (http://compbio.dfci.harvard.edu/tgi/plant.html). Based on the reduplicated assembled contigs, we synthesized multiple primers, which were used for amplifying the cDNA of *HvG6PDHs* (Appendix A). Multiple alignments of DNA and protein sequences were conducted using CLUSTAL X. The transit peptide sequences of HvG6PDH proteins were evaluated using the ProtComp 9.0 software (Softberry, http://linux1.softberry.com/).

A phylogenetic tree was constructed through a multiple alignment of amino acid sequences from *T. urartu* (EMS47668, EMS45717, EMS62742), *A. tauschii* (XP_020168771, XP_020156901, EMT03892, EMT16963, EMT23941), *B. distachyon* (XP_003580004, XP_003575296, XP_003558039, XP_003581458), *O. sativa* (XP_015627059, AAL79959, ABF95637, BAC84352, ABF96582), *Z. mays* (AFW57831, DAA45780, ACG39996, XP_008658752), *S. bicolor* (XP_002446185, XP_021306709, XP_021313884, XP_021318732), *A. thaliana* (Q9FJI5, Q43727, Q9IK23, Q8I743, Q93ZW0, Q9FY99), *P. trichocarpa* (Q9FY99, Q9FY99, EEE79649, ERP53365), *R. communis* (EEF50009, EEF47431, EEF32168), and *H. vulgare* using the MEGA 7.0 software (http://www.megasoftware.net/) with the Neighbor-Joining method. The robustness of each node in the phylogenetic tree was evaluated with 1000 replicates.

### 2.3. RNA Extraction and Quantitative Real-Time PCR (qRT-PCR)

Total RNA was extracted from highland barley leaves or roots using Trizol reagent (Sangon, Shanghai, China). The cDNA synthesis was carried out using the M-MLV Reverse Transcriptase Kit (Sangon, Shanghai, China). After that the cDNA was diluted with RNAase-free water, which was used to assess the expression of *HvG6PDH* genes by qRT-PCR. The primers for *HvG6PDH* and *Hv**ACTIN* (accession number: MK034133) genes are listed in Appendix A.

qRT-PCR was performed in a PCR cycle system (ABI 7500, ThermoFisher Scientific, USA) using the SYBR Green PCR master mix (Takara, Dalian, China). The PCR reaction was performed using the program of 95 °C for 30 s, and 40 cycles of 95 °C for 5 s and 60 °C for 34 s. The gene expression level was calculated according to the 2^−ΔΔCt^ method [45], where the ^ΔΔ^Ct value was the Ct value of *HvG6PDH* genes minus the Ct value of the *Hv**ACTIN*.

### 2.4. Subcellular Localization

The open reading frame (ORF) of *HvG6PDH1* to *HvG6PDH5* were cloned into a pENTR^TM^/D-TOPO^®^ vector (Invitrogen, USA) to generate *att*L1 and *att*L2 fragments into 5′ and 3′ terminal of *HvG6PDH1* to *HvG6PDH5* ORFs, respectively. The entry vectors (pENTR-*HvG6PDH1* to pENTR-*HvG6PDH5*) were recombined into the *att*R1 and *att*R2-containing expression vector PGWB-RFP to produce HvG6PDHs-RFP proteins, which were driven by 35S promoter using the Gateway technology. All primers in the experiment are listed in Appendix A. All plasmids of destination constructs were transformed into the *A. tumefaciens* GV3101, and the resulting positive clones were infiltrated into *N. benthamiana* leaf for 48 h for transient expression [46]. The transformed *N. benthamiana* leaves were visualized through a laser scanning confocal microscope (C2 Plus, Nikon, Japan). The excitation wavelength for RFP and chloroplast autofluorescence was 561 nm and 556 nm, respectively, and the emission wavelength was 595 nm and 650 nm, respectively.

### 2.5. SDS-PAGE of HvG6PDH Proteins from Escherichia coli

Five *HvG6PDHs* ORFs were cloned into the vector pET28a(+), and then were transformed into *E. coli* BL21. Positive clones were inoculated into fresh liquid LB medium with 50 µg mL^−1^ kanamycin and cultured until the OD_600_ reached 0.5–0.6. Then 1.0 mM isopropyl b-D-1-thiogalactopyranoside (IPTG) was added and induced protein expression. After shaking for 2 h, 10 mL LB medium was boiled for 7 min at 95 °C, and then centrifuged at 10,000× *g* for 10 min. After centrifugation, 10 mL supernatant was collected and used for protein electrophoresis in 12% SDS-PAGE. For analysis of the HvG6PDHs enzymes, the induced strains were ground but not boiled. The empty vector PET28(a) was used as the control. The SDS-PAGE gel was stained with Coomassie Brilliant Blue dye solution.

### 2.6. Extraction and Determination of G6PDH Activity

The extraction and assay of G6PDH were determined according to Hauschild and von Schaewen [47]. In order to determine the cytosolic G6PDH activity, the samples were incubated with 62.5 mM DTT at 25 °C for 5 min before measuring the enzyme activity, which can result in the inactivation of plastidic G6PDH [48]. Plastidic G6PDH activity was the difference between total and cytosolic G6PDH. In the analysis of enzyme activity, the extraction buffer was added into the reaction solution instead of the crude enzyme extract as blank control. Protein contents were analyzed using the method of Bradford [49].

### 2.7. Determination of Hydrogen Peroxide and Superoxide Anion Contents

Hydrogen peroxide (H_2_O_2_) content was measured as described by Veljovic-Jovanovic et al. [50]. H_2_O_2_ content was estimated by measuring the absorbance at 590 nm and determined from a calibration curve using 10–50 μM H_2_O_2_ as the standard. Superoxide anion (O_2_^−^) content was determined according to Elstner and Heupel [51]. Briefly, O_2_^−^ was estimated by measuring the absorbance at 530 nm and recorded in a UV-VIS spectrophotometer.

### 2.8. Statistical Analysis

Each experiment was repeated at least three times. Data were expressed as mean ± SE, and analyzed by one-way variance analysis (ANOVA).

## 3. Results

### 3.1. Identification and Preliminary Characterization of Putative HvG6PDH Genes in Highland Barley

Highland barley, also named naked barley, is a relative species of barley (*Hordeum vulgare* L). In the study, five putative *G6PDH* genes in the genome of barley were selected as candidates to clone and identify *HvG6PDH* genes from the cDNA database of highland barley, which were finally named as *HvG6PDH1*, *HvG6PDH**2*, *HvG6PDH**3*, *HvG6PDH**4*, *HvG6PDH5*. The cDNA sequences of *HvG6PDH**1-5* genes were submitted to GeneBank with accession numbers listed in Table 1. The number of *G6PDH* genes in highland barley is similar to the number reported in other species: six in *A. thaliana* [49], five in *O. sativa*, four in *T. urartu* [52], four in *Z. mays* [53], and five in *A. tauschii* [54].

G6PDH isozymes have various biochemical properties in plants [28,55,56]. In order to analyze the biochemical properties of HvG6PDH proteins, we predicted the isoelectric point (pI), molecular weight, and subcellular localization of each HvG6PDH protein. The open reading frame (ORF) length and deduced amino acid number of *HvG6PDH1* to *HvG6PDH5* range from 1527 to 1788 bp and 509 to 596 amino acids, respectively. The molecular weight and pI values range from 57.9 to 67.0 kD and 5.4 to 8.3, respectively (Table 1). Analysis of predicted transit peptides and subcellular localization of the HvG6PDH proteins indicated that HvG6PDH2 and 5 are localized in the cytosol, while HvG6PDH1, 3, and 4 are localized in the chloroplast/plastid (Table 1). A multiple comparison analysis of HvG6PDH proteins showed that the identity of amino acid sequences ranges from 44.8% to 81.6%, and the highest and lowest identity was found between HvG6PDH2 and 5 and between HvG6PDH1 and 3, respectively. Three highly conserved amino acid sequences were found in the substrate-binding site (IDHYLG), NADP-binding site (NEFVIRLQP), and Rossman fold (GASGDLAKKK) domain (Figure 1).

To confirm whether the encoded proteins of these putative *HvG6PDH* genes possess G6PDH activity, we expressed *HvG6PDH* genes in the pET28(a) vector driven by a T7 promoter. The results suggested that the five *HvG6PDH* genes are highly expressed with the expected molecular mass (Appendix A). The expressed HvG6PDH1 to HvG6PDH5 proteins showed the remarkable G6PDH activity in comparison with that in the empty vector (the own G6PDH activity of *E. coli.*). These results indicated that the five *HvG6PDH1-5* are members of the *G6PDH* gene family in highland barley, and encode the functional G6PDH proteins.

### 3.2. Phylogenetic Profiling and Subcellular Localization of HvG6PDH Proteins

To explore the similarity between HvG6PDHs and other determined G6PDH proteins, we constructed a phylogenetic tree based on a multiple alignment of HvG6PDHs and other 43 G6PDH proteins from monocots (*Z. mays*, *T. urartu*, *S. bicolor*, *A. tauschii*, *O. sativa*, *T. aestivum*, and *B. distachyon*) and eudicots (*P. trichocarpa*, *R. communis*, and *A. thaliana*; Figure 2). Amino acid alignment revealed that both NADP^+^ and glucose-6-phosphate binding motifs are highly conserved in these G6PDH proteins (Figure 1). Based on the classification of G6PDH proteins in *A. thaliana*, G6PDH1 to G6PDH4 has been verified to locate in plastids, and G6PDH5 and G6PDH6 are located in the cytoplasm [14,21]. Four clusters are formed in the phylogenetic cladogram. The early branch corresponding to plastidic G6PDH isoforms is further divided into three subclusters (P0, P1, and P2), in which HvG6PDH1, 3 and 4, are distributed. The cytosolic G6PDH (Cyt) branch includes HvG6PDH2 and HvG6PDH5, (Figure 2). It is worth noting that the HvG6PDH isoforms share high homology with those G6PDHs from *T. aestivum*, *T. urartu*, and *A. tauschii* in each cluster, with 85%-95% homology in amino acids among the G6PDH members (Figure 2). To further ascertain the subcellular localization of HvG6PDH1 to HvG6PDH5, we fused *HvG6PDHs* into the N-terminal of GFP and transiently expressed it in *N. benthamiana* leaves. As expected, HvG6PDH1, 3, 4 are unequivocally located in chloroplasts, and HvG6PDH2 and HvG6PDH5 proteins are clearly located in the cytoplasm (Figure 3), confirming the results from phylogenic analysis and previous online prediction for HvG6PDHs.

### 3.3. NaCl and PEG Treatments Induce the G6PDH Activity in Highland Barley

To investigate the responses of G6PDH under NaCl and PEG conditions, we determined the activities of cytosolic and plastidic G6PDH in leaves and roots from highland barley seedlings. In the presence of 150 mM NaCl, the cytosolic G6PDH activity in leaves was significantly increased at 120 h and 144 h, but the plastidic G6PDH activity did not change (Figure 4A). The cytosolic and plastidic G6PDH activities in roots reached their maximum under NaCl treatment for 96 h, and were increased to 4.5-fold and 1.7-fold of control, respectively (Figure 4B). In contrast, PEG treatment rapidly induced the activity of cytosolic G6PDH and the highest value in leaves (2.0-fold) and roots (2.5-fold) appeared at 120 h and 96 h, respectively (Figure 4C,D). The activity of plastidic G6PDH in roots had a remarkable increase from 72 h to 120 h but had no obvious change in leaves (Figure 4C,D). Moreover, G6PDH activity was similar among the seedlings at different growth stages under the control conditions (Appendix A) These results suggested that NaCl and PEG treatment can induce the activity of G6PDH in highland barley, and the cytosolic G6PDH is more responsive than plastidic G6PDH in salt and drought stresses.

### 3.4. NaCl and PEG Treatments Activate the Expression of HvG6PDH Genes in Highland Barley

To further understand the response of *HvG6PDH* genes under NaCl and PEG treatments, we examined the expression level of five *HvG6PDH* genes in the leaves and roots of highland barley seedlings. Under normal conditions, the five *HvG6PDH* genes exhibited different expression levels in roots and leaves (Figure 5). The transcript level of *HvG6PDH5* in leaves upon 150 mM NaCl stress showed no change within 120 h, whereas the expression of *HvG6PDH1*, *HvG6PDH2*, *HvG6PDH3,* and *HvG6PDH4* rapidly increased after 3 h and reached peak values at 12, 6, 12, and 12 h, respectively (Figure 5A). However, when highland barley roots were exposed to NaCl treatment, the transcript level of *HvG6PDH1*, *HvG6PDH2*, and *HvG6PDH4* decreased sharply after 3 h, but gradually increased after 36 h. The expression of *G6PDH3* and *G6PDH5* had no obvious changes in this period (Figure 5B).

Figure 5C showed that under 20% PEG treatment, except for Hv*G6PDH3* and Hv*G6PDH**4* expression, the *HvG6PDH**1, HvG6PDH**2, HvG6PDH**5* transcription gradually increased after 3 h and reached the highest values during 12–36 h compared to those in the leaves of the untreated seedlings (0 h). In roots, the expression of *HvG6PDH2* peaked at 72 h under 20% PEG treatment; *HvG6PDH1* and *HvG6PDH4* gradually increased and decreased during the treatment, respectively; comparatively, *HvG6PDH3* had no change (Figure 5D). These results suggest that the expression patterns of five *HvG6PDH* genes vary in response to salt and drought stresses, implying that they may play different roles in these processes.

### 3.5. Abscisic Acid and Jasmonic Acid Positively Regulate the Activity and Expression of G6PDH

Abscisic acid (ABA) and jasmonic acid (JA) are critical signaling molecules in regulating gene expression under salt and drought stresses [57]. To determine whether ABA and JA affect G6PDH in highland barley, we analyzed the activity of cytosolic and plastidic G6PDH and the transcript abundance of *HvG6PDHs* under ABA and JA treatments. In the presence of ABA, the activity of cytosolic and plastidic G6PDH in leaves was rapidly increased after 6 h and reached the highest value at 12 h, and which increased by 47% and 113% compared to the control, respectively (Figure 6A). Moreover in roots, the activity of cytosolic G6PDH showed a similar trend with that in leaves, but the plastidic G6PDH activity gradually increased during the treatment (Figure 6B). Under JA treatment, the activity of both cytosolic and plastidic G6PDH in leaves reached the maximum at 24 h and then displayed a decreasing trend (Figure 6C). However, in roots, they reached the highest value at 12 h. The activity of cytosolic and plastidic G6PDH increased by 22% and 66%, respectively (Figure 6D).

The CGTCA motif (JA-responsive element) was found in the promoters of *HvG6PDH1*, *HvG6PDH2*, *HvG6PDH3,* and *HvG6PDH4*, and the ABRE motif (ABA-responsive element) was also found in the promoters of *HvG6PDH2*, *HvG6PDH3*, and *HvG6PDH4*. Neither CGTCA nor ABRE motif was found in the promoter of *HvG6PDH5* (Appendix A). As expected, under ABA treatments, the expression levels of *HvG6PDH2*, *HvG6PDH3,* and *HvG6PDH4* in leaves increased rapidly after 3 h and reached the highest values at 12 h. The expression levels of *HvG6PDH1* and *HvG6PDH5* had no changes within 120 h of ABA treatment (Figure 7A). In roots, these genes showed similar patterns of expression as in leaves under ABA treatment, but they peaked at 6 h (Figure 7B). JA treatment also can stimulate the expression of *HvG6PDH**3* and *HvG6PDH4* in leaves (Figure 7C) and the expression of *HvG6PDH**1*, *HvG6PDH**2*, *HvG6PDH**3,* and *HvG6PDH4* in roots (Figure 7D).

### 3.6. G6PDH Is Involved in ROS Scavenging under Salt and Drought Stresses

To elucidate the role of G6PDH in salt and drought stresses, Glucm, a competitive inhibitor of G6PDH [58], was used in this study. As expected, the activity of plastidic G6PDH was significantly decreased, while the activity of cytosolic G6PDH had a slight decline in both leaves and roots under Glucm treatment alone (Appendix A). Appendix A showed that the activities of cytosolic and plastidic G6PDH were obviously stimulated by 150 mM NaCl or 10% PEG treatment; and the 200 mM NaCl or 20% PEG treatment induced the higher increase of their activities in the leaves and roots of highland barley. Application of Glucm (10 mM) could markedly inhibit the G6PDH activities induced by NaCl and PEG treatments (Appendix A).

G6PDH is the main source of NADPH production and NADPH regulates the balance of ROS level in animals and yeast [59,60]. To declare G6PDH function in ROS production in highland barley, we examined the effect of Glucm on H_2_O_2_ and O_2_^−^ contents under salt and drought stresses. Compared to the control, the application of Glucm alone had almost no effects on H_2_O_2_ and O_2_^−^ contents (Figure 8). Salt and drought stress significantly increased the H_2_O_2_ and O_2_^−^ contents in highland barley (Figure 8). For example, under 200 mM NaCl treatment, the H_2_O_2_ and O_2_^−^ contents in leaves increased to 2.2-fold and 1.6-fold, respectively; while in roots they increased to 2.5-fold and 2.2-fold, respectively. Similarly, under 20% PEG treatment, the H_2_O_2_ and O_2_^−^ contents in leaves were increased to 1.9-fold and 1.4-fold, respectively; in roots, they were increased to 2.4-fold and 2.1-fold, respectively (Figure 8). The application of Glucm under NaCl or PEG treatment further increased the H_2_O_2_ and O_2_^−^ contents in comparison with the NaCl treatment alone or PEG treatment alone in highland barley, respectively (Figure 8). The results suggested that G6PDH probably has a critical role in ROS removal in highland barley.

## 4. Discussion

In this study, we first identified five *HvG6PDH* genes in highland barley, and their sequences were confirmed with the published highland barley genome sketch [38]. The substrate-binding (IDHYLG) and NADP-binding (NEFVIRLQP) motifs are necessary for G6PDH activity [14,56]. Indeed, we found that the five deduced HvG6PDH protein sequences contain both “IDHYLG” and “NEFVIRLQP” motifs (Figure 1), indicating the corresponding G6PDH proteins should be active. Expression of the five *HvG6PDH* genes in *E. coli* indicated that these genes encode the active G6PDH proteins (Appendix A). The online software and subcellular localization analyses revealed that HvG6PDH2 and HvG6PDH5 are localized in the cytosol, while the other three HvG6PDHs (HvG6PDH1, 2 and 4) belong to the plastidic isoforms (Table 1; Figure 3). The cytosolic and plastidic HvG6PDH isoforms are consistent with reported orthologues in *A*. *thaliana*, *O*. *sativa*, *T*. *urartu*, *Z*. *mays,* and *H. brasiliensis* [14,52,53,54,56]. Moreover, the numbers of cytosolic and plastidic G6PDH isoforms are different in the diverse plant species. For example, plastidic G6PDH isoforms in *A*. *thaliana*, *O*. *sativa*, *T*. *urartu*, *Z*. *mays*, and *B. distachyon* are 4, 3, 3, 3, and 2, respectively, which is probably associated with the protein functional redundancy.

As a reductive coenzyme, NADPH not only provides the reduced power for the regeneration of reduced glutathione (GSH) but is the substrate of the plasma membrane NADPH oxidase to participate in H_2_O_2_ production [44,61,62]. NADPH is originated from photosynthesis and the OPPP pathway. In non-photosynthetic tissues, NADPH production is sourced through the OPPP [13]. Numerous literature have reported that G6PDH is widely induced in aluminum, drought, salt, and low-temperature stresses [23,24,25,26,27,28,29,30,33,34]. Under salinity or drought conditions, the activity of cytosolic and plastidic G6PDH significantly increases in highland barley seedlings, and the cytosolic G6PDH is more sensitive than the plastidic G6PDH (Figure 4). Liu et al. reported that G6PDH regulates H_2_O_2_ metabolism by maintaining GSH and ASC contents under drought in soybean [23]. Moreover, it was thought that the NO-modulated cytosolic G6PDH is responsible for ROS accumulation through the PM NADPH oxidase, resulting in the inhibition of soybean root elongation under high aluminum stress [31]. The results suggest cytosolic G6PDH should be involved in the regulation of the cellular redox homeostasis by supplying NADPH against abiotic stress.

In plants, the cytosolic and plastidic *G6PDH* genes have different expression patterns, and their gene expression does not positively correlate with the enzymatic activity [21,56]. Thus, the G6PDH isoforms should be regulated at the transcription and translation level. In the study, gene expression analysis indicated the mRNA abundance of cytosolic *HvG6PDH2* is significantly increased, but *HvG6PDH5* does not change in highland barley under salt and drought stresses (Figure 5). The significant enhancement in cytosolic *G6PDH* genes during salinity or drought conditions has also been reported in Para rubber tree and sugarcane [34,56]. Moreover, overexpression of cytosolic *PsG6PDH* from poplar confers cold tolerance in transgenic tobacco [33]. The *Saccharomyces cerevisiae* overexpressing *Cv**G6PDH* from *Chlorella vulgaris* increased the tolerance to cold [63]. Although the plastidic *HvG6PDH* genes (*HvG6PDH1* and *4*) were largely induced, the activity of the plastidic enzyme had no change under salt and drought stresses (Figure 4 and Figure 5). These results imply that HvG6PDH2, as a cytosolic G6PDH, mainly participated in the tolerance to abiotic stresses.

ABA and JA are widely involved in plant response to stresses, such as salinity, drought, cold, wounding, and so on [57,64,65]. ABA response elements (ABREs) and CGTCA motif have been identified in the promoters of ABA and JA-induced genes, respectively [66,67]. In this study, we found that promoters of *HvG6PDH1*, *HvG6PDH2*, *HvG6PDH3,* and *HvG6PDH4* have the CGTCA motif and the promoters of *HvG6PDH2*-*HvG6PDH4* have ABRE motif. *HvG6PDH5* has neither CGTCA nor ABRE motif (Appendix A). As expected, ABA treatment induced the expression of *HvG6PDH2*, *HvG6PDH3,* and *HvG6PDH4*, but *HvG6PDH1* and *HvG6PDH5* had no responses to ABA treatments (Figure 7). Moreover, the activities of cytosolic and plastidic G6PDH were significantly increased in highland barley under ABA treatment (Figure 6A,B). ABRE motif also has been found in the promoters of *P2-G6PDH* in rice and barley [68]. Cardi et al. reported that ABA can activate the activity of the P2-isoform G6PDH, but not the cytosolic G6PDH in barley [28]. However, the down-regulation of *HbG6PDHs* transcription was also found in latex from *H. brasiliensis* after ABA treatment [56]. In addition to ABA, the cytosolic and plastidic G6PDH activities were significantly increased in highland barley exposed to JA treatment (Figure 6C,D). Moreover, except for *HvG6PDH5*, the expression level of all *HvG6PDHs* was obviously induced by JA treatment in both leaves and roots (Figure 7C, D). Long et al. found that JA treatment activated the gene expression of cytosolic and plastidic *G6PDH* [56]. These results imply that the cytosolic and plastidic G6PDH play an important role in ABA and JA responses in highland barley seedlings.

With the environmental changes, plant cells need to alter the cellular metabolism to adapt to stresses. One of these metabolic changes is in the ROS level, which is a universal phenomenon in plant response to environmental stresses [2,3,9,10]. At a low level, ROS can act as the signaling molecules, but at an excessive level, it can lead to oxidative damages [2,10,19,69]. G6PDH can supply massive NADPH to regulate the ROS level [20,47,61,70]. Moreover, G6PDH is also required for plant development by regulating carbon and nitrogen flow [21,56].

In animals and yeast, NADPH is mainly produced through the OPPP to maintain the ROS homeostasis [59,60]. Here, we found that Glucm (a competitive inhibitor of G6PDH) treatment significantly decreased the activity of cytosolic and plastidic G6PDH (Appendix A), accompanied by the high ROS contents (Figure 8) in highland barley under salt and drought stresses. Dal Santo et al. assumed that photosynthesis could provide reducing equivalents in plants [2]. When plants are exposed to unfavorable conditions, photosynthesis may be impaired. In this case, the reducing power for the regulation of intracellular redox balance may be mainly provided by OPPP. The GSH production is impaired in the *At**g6pd6* mutant, leading to enhanced sensitivity to salt stress in Arabidopsis [2]. Scharte et al. reported that overexpression of a plastidic *G6PD* gene in tobacco enhanced the tolerance to drought [58]. When the G6PDH activity is disrupted, mouse embryonic stem cells are highly sensitive to oxidative stress [59,71]. Similarly, G6PDH-defective yeast mutants are very susceptible to oxidative stress [60,72]. These results, together with our observation in highland barley, suggest that G6PDH, a key enzyme of OPPP, plays a pivotal role in maintaining the cellular redox balance in stress acclimation.

In summary, G6PDH is a primary regulator of ROS removal and plays an important role in plant acclimatization to stresses. We cloned five *HvG6PDH* genes from highland barley and confirmed that *HvG6PDH1**-HvG6PDH4* are involved in response to salt and drought stresses. Among them, the cytosolic HvG6PDH2 is the major isoform against oxidative stress. Moreover, JA and ABA are probably critical regulators of *HvG6PDH*s (except *HvG6PDH5*) expression under salt and drought stresses.

## Figures and Tables

**Figure 1 plants-09-01800-f001:**
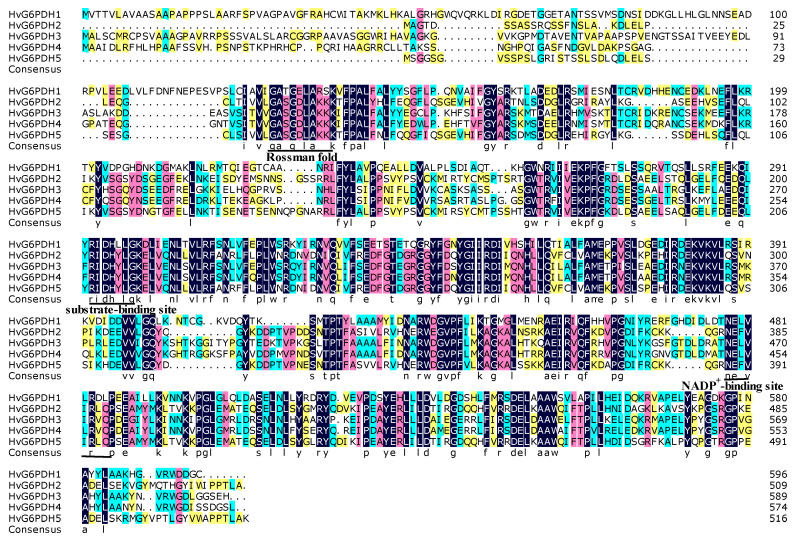
Amino acid sequence alignment of the five HvG6PDH isoforms from highland barley. The strictly conserved and different amino acid residues were indicated with black and white backgrounds, respectively. Green and red shades show the conserved amino acid residues with ≥75% and ≥50%, respectively. The three putative conserved sequences with (GA × G × LA), (RIGH × LG), and (NE × VR × P) of HvG6PDH isoforms are underlined and labeled as Rossman fold, substrate binding site, and NADP^+^ binding site, respectively. The sequences were aligned using the DNAMAN 6.0 software (Lynnon Biosoft, USA). The numbers on the right indicate amino acid positions.

**Figure 2 plants-09-01800-f002:**
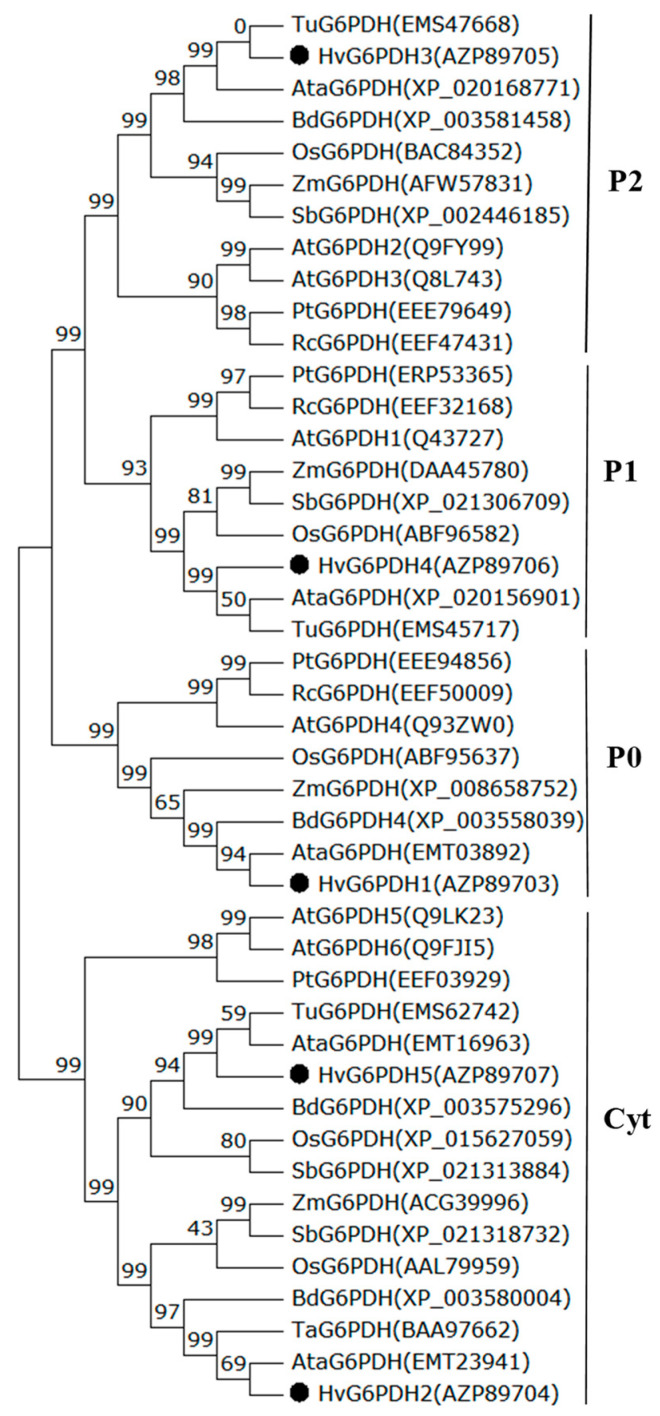
Phylogenetic profiling of plant G6PDH isoforms. A distance cladogram using the Neighbour-Joining method shows the majority consensus of the G6PDH proteins from highland barley (*Hordeum vulgare*), *Oryza sativa* (Os), *Zea mays* (Zm), *T. aestivum* (Ta), *Triticum urartu* (Tu), *Aegilops tauschii* (Ata), *Brachypodium distachyon* (Bd), *Sorghum bicolor* (Sb), *Populus trichocarpa* (Pt), *Ricinus communis* (Rc), and *Arabidopsis thaliana* (At). The four G6PDH clades are indicated and clades P (0, 1, 2) and Cyt correspond to plastidic and cytosolic G6PDH isoforms, respectively. Bootstrap values are indicated at the branch points. Barley sequences are marked in bold dots.

**Figure 3 plants-09-01800-f003:**
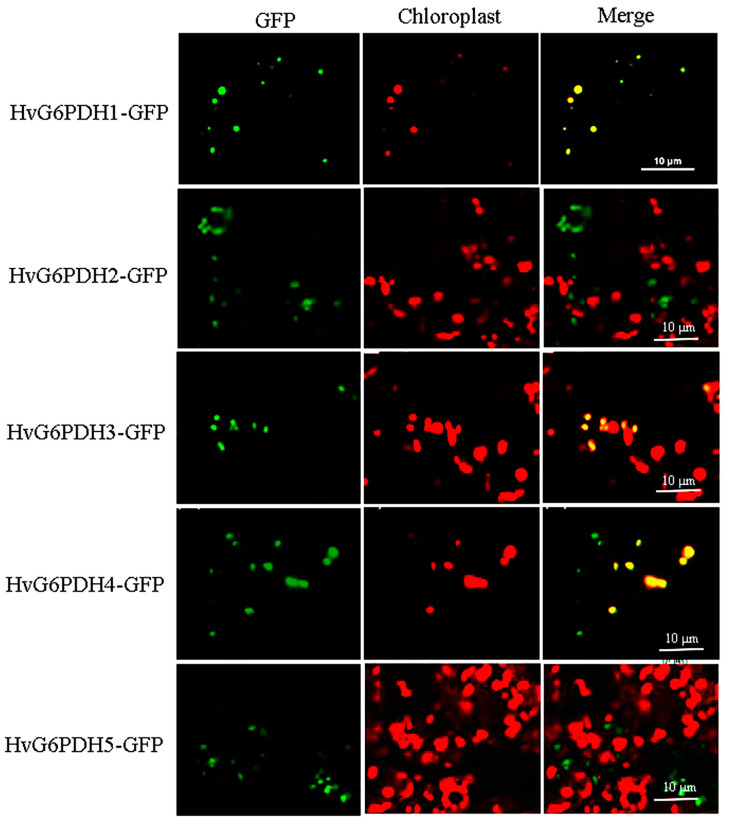
Subcellular localization of HvG6PDHs-GFP in *N. benthamiana* leaves. The photographs were taken after transient expression of *HvG6PDHs-GFP* genes for 48 h. Scale bar = 10 µm. Green indicates the fluorescence of HvG6PDHs-GFP proteins. Red and yellow show the chloroplast autofluorescence and merged GFP and chloroplast colors, respectively.

**Figure 4 plants-09-01800-f004:**
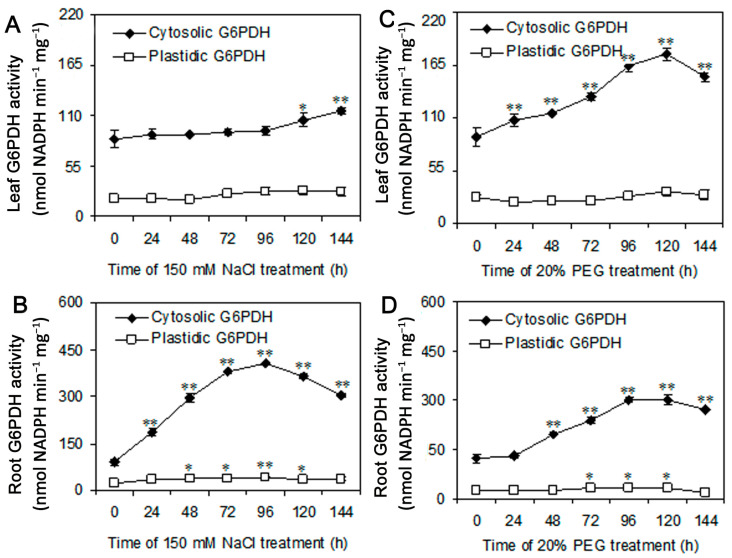
The activity of cytosolic and plastidic G6PDH in leaves and roots of highland barley under NaCl (**A**,**B**) and PEG (**C**,**D**) treatments for 0 h to 144 h. Data are mean ± SE (n = 3). Statistical differences between Control (0 h, the untreated seedlings) and NaCl/PEG treatment at every time point were analyzed on the basis of Student’s t-test (*p* < 0.05 * and *p* < 0.01 **).

**Figure 5 plants-09-01800-f005:**
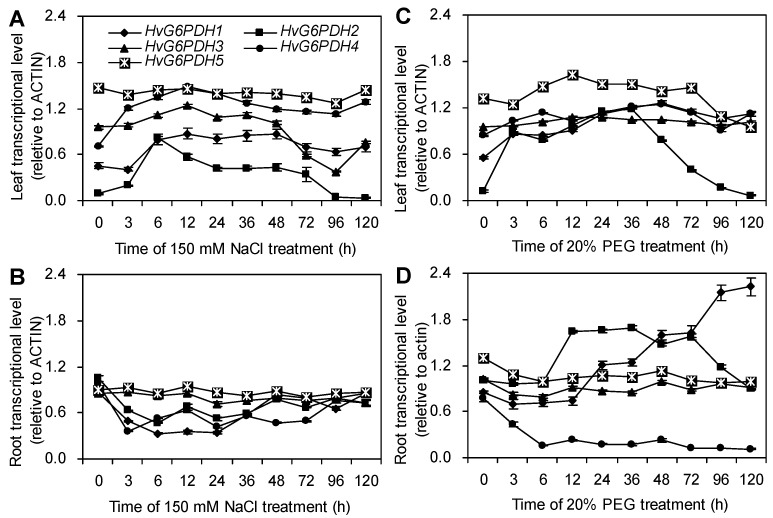
Time courses of *HvG6PDH**s* gene expression in leaves and roots of highland barley under NaCl (**A**,**B**) and PEG (**C**,**D**) treatments for 0–120 h. The relative expression level of *HvG6PDH**s* was normalized to *Hv**ACTIN*, and then they were expressed as a ratio of the value of the untreated seedlings (0 h) for each treatment (*n* = 3).

**Figure 6 plants-09-01800-f006:**
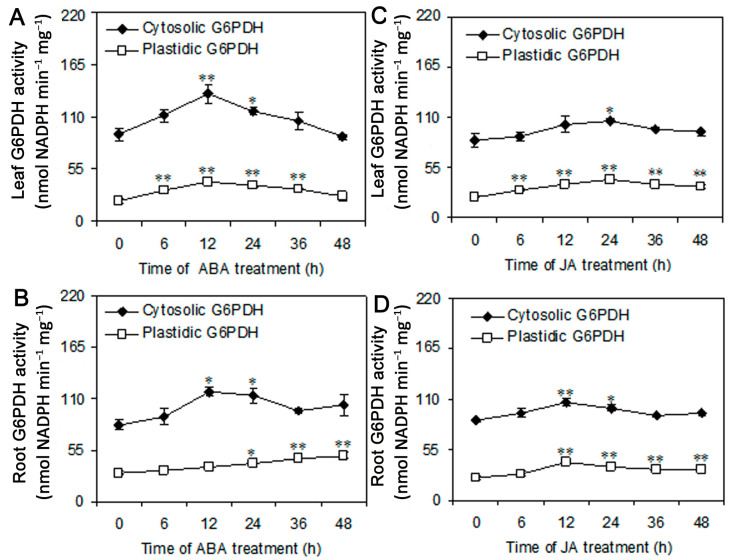
The activity of cytosolic and plastidic G6PDH in leaves and roots of highland barley under abscisic acid (ABA) (**A**,**B**) and jasmonic acid (JA) (**C**,**D**) treatment for 48 h. Data are mean ± *SE* (n = 3). Statistical differences between Control (0 h, the untreated seedlings) and ABA/ JA treatment at every time point were analyzed on the basis of Student’s t-test (*p* < 0.05 * and *p* < 0.01 **).

**Figure 7 plants-09-01800-f007:**
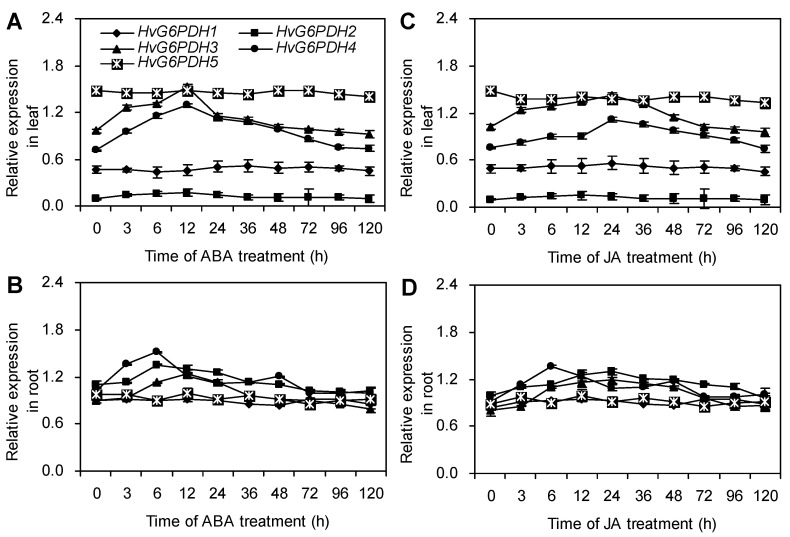
Time courses of *HvG6PDH**s* gene expression in leaves and roots of highland barley under ABA (**A**,**B**) and JA (**C**,**D**) treatments. The relative expression levels of *HvG6PDH**s* were normalized to *Hv**ACTIN*, and then they were expressed as a ratio of the value of the untreated seedlings (0 h) for each treatment (n = 3).

**Figure 8 plants-09-01800-f008:**
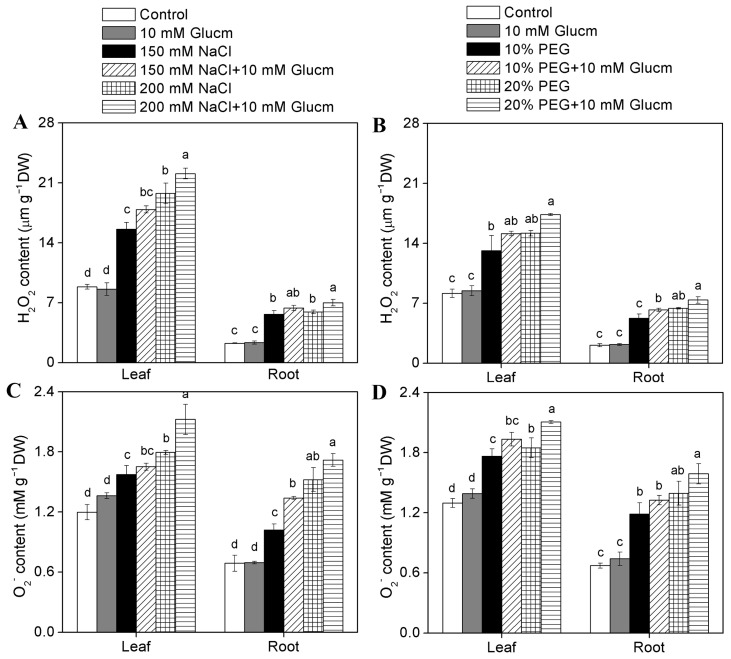
Effect of G6PDH inhibitor (Glucm) on H_2_O_2_ and O_2_^−^ contents in leaves and roots of highland barley under NaCl (**A**,**B**) and PEG (**C**,**D**) treatments for 48 h. Data are mean ± *SE* (n = 3). Statistical differences were analyzed on the basis of the Student’s *t*-test, and bars with different letters were different at the 0.05 level.

**Table 1 plants-09-01800-t001:** Protein properties of HvG6PDHs. The subcellular location of HvG6PDHs was predicted using WoLF PSORT (https://www.genscript.com/psort.html). The protein sequences encoded by *HvG6PDH* genes were predicted for amino acids number (aa), transit peptide length (TP), isoelectric point (pI), and molecular weight (MW) using the protein tools (properties) in GENE INFINITY website (http://www.geneinfinity.org/index.html?dp=5). The location of *HvG6PDH* genes is based on barley chromosome sequence.

Gene Name	Chromosome Location	Predicted Localization	aa	TP	pI	MW (kD)	GenBank
***HvG6PDH1***	4H:433572159-433576826	Chloroplast	596	27	5.4	67.0	MK034128
***HvG6PDH2***	2H:608625462-608631046	Cytosol	509	−	6.3	57.9	MK034129
***HvG6PDH3***	2H:20076557-20081131	Chloroplast	589	39	8.3	65.6	MK034130
***HvG6PDH4***	6H:114688120-114690749	Chloroplast	574	37	8.2	64.8	MK034131
***HvG6PDH5***	6H:342276577-342281388	Cytosol	516	−	5.9	58.3	MK034132

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
