# Peer review of "Identification, Characterization, and Stress Responsiveness of Glucose-6-phosphate Dehydrogenase Genes in Highland Barley"

_plants, 2020, doi:10.3390/plants9121800_

Round 1

Reviewer 1 Report

I went through the manuscript to make sure that my suggestions for adjustments were accepted. Some adjustments were done, others not:

  1. Abstract: It is necessary to explain the abbreviation for NADPH? (still explained)
  2. Introduction: Citations of involvement of G6PDH in stress: please indicate that there are not presented all published citations or add others. (done)
  3. Introduction: day/night regulation of G6PDH is not mentioned (done)
  4. Methods: Please indicate: the number of biological repetitions and number of plants in each abiotic stress experiment (done)
  5. Methods: What is the principle of the separation of G6PDH activity to cytosolic and plastidic isoform using DTT? It is reduction of disulfide bonds of transit peptide? Does it coincide with the separation by centrifugation in a sucrose gradient? (-)
  6. 4 Does the abbreviation in the unit of enzyme activity Pro correspond to the protein? Then I recommend title specific activity [nmol NADPH.min-1.mg-1] (not accepted)
  7. Results: p. 7 l. 232, 236 and p. 11, l. 313, 314, 315 If the -fold is lower than 1, it is not increase..e.g. They increased by 0.4-fold.. What does it mean? From 1 to 0.4, it is not increase.. (done)
  8. Discussion: The site of action of the inhibitor should be specified (in the watering treatment, or whether it was added only during the measurement) (not specified)

As these were not serious comments, I recommend accept the MS after minor revisions. 

Author Response

Dear Reviewer;

Thank you so much for your constructive comments on our manuscript The manuscript have been revised carefully.

>Reviewer 1

  1. Abstract: It is necessary to explain the abbreviation for NADPH? (still explained)

    >Thanks, it has been revised (L14-15)

  1. Introduction: Citations of involvement of G6PDH in stress: please indicate that there are not presented all published citations or add others. (done)

    >Thanks, it has been added (L49-51).

  1. Introduction: day/night regulation of G6PDH is not mentioned (done)

>Thanks, it has been added (L51-52).

  1. Methods: Please indicate: the number of biological repetitions and number of plants in each abiotic stress experiment (done)

>Thanks, it has been added (L164, 261-263, 286-287, 304, 321-322, 348).

  1. Methods: What is the principle of the separation of G6PDH activity to cytosolic and plastidic isoform using DTT? It is reduction of disulfide bonds of transit peptide? Does it coincide with the separation by centrifugation in a sucrose gradient? (-)

       > Thanks for your constructed advice. Plastidic G6PDH has two states: oxidation and reduction, and the oxidation state is its active form. Wenderoth et al. (1997) reported that Cys residue is the redox regulatory site of Plastidic G6PDH. When there is enough DTT in the reaction solution, it could be reduced. And in the revised MS, we added the related information (152-153).

Wenderoth I, Scheibe R, von Schaewen A (1997) Identification of the cysteine residues involved in redox modification of plant plastidic glucose-6-phosphate dehydrogenase. J Biol Chem 272: 26985-26989

  1. Does the abbreviation in the unit of enzyme activity Pro correspond to the protein? Then I recommend title specific activity [nmol NADPH.min-1.mg-1] (not accepted)

       >Thanks. I'm sorry, and I misunderstood your suggestion at first. And in the current revision, the enzyme activity unit has been corrected (Figs. 4, 6, S1, S4).

  1. Results: p. 7 l. 232, 236 and p. 11, l. 313, 314, 315 If the -fold is lower than 1, it is not increase..e.g. They increased by 0.4-fold.. What does it mean? From 1 to 0.4, it is not increase.. (done)

    >Thanks for your constructed advice, they have been revised.

  1. Discussion: The site of action of the inhibitor should be specified (in the watering treatment, or whether it was added only during the measurement) (not specified)

 >Thanks for your advice, we added and revised the related description (88-91).

As these were not serious comments, I recommend accept the MS after minor revisions. 

Reviewer 2 Report

I just realized that I have reviewed this manuscript three times before and it came back as resubmission. Still I do not see any suggested changes incorporated. Attached is the first review report. I think it is good to get other reviewers' opinions on this.

Author Response

Dear Reviewer;

Thank you so much for your constructive comments on our manuscript The manuscript have been revised carefully.

>Reviewer 2

>I just realized that I have reviewed this manuscript three times before and it came back as resubmission. Still I do not see any suggested changes incorporated. Attached is the first review report. I think it is good to get other reviewers' opinions on this.

>We are very sorry that our modification did not meet your requirements. And in the present revision, we have carefully analyzed your suggestions and made corresponding modifications.

The main changes are as follows:

>1. Page 3, line 10: How cDNA was dissolved in DEPC water and adjusted to 50-100nM

concentration?

>2. Page 3, line 10: Please check DEPC water concentration (1% or 0.1%).

 >Thanks for your constructive advice. I'm very sorry. These are a mistake! And they has been revised (L118-119).

>3. Manuscript needs professional English editing. For example:

Page 2, line 66: Change “responsing” to “response”.

Thanks, it has been revised (L70)

Page 2, line 89: Remove “literately”.

Thanks, it has been deleted.

Page 3, line 128: Change “leaves” to “leaf”. 134

Thanks, it has been revised (L134)

Page 3, line 132: Remove “Subcellular localization”.

Thanks, it has been deleted.

Reviewer 3 Report

The manuscript has improved a lot after the corrections! Adding the supplementary figure 2, which demonstrates that the enzyme activity does not change with the age of the plants in leaves and roots, helps a lot, because it supports the conclusions made about changes in enzyme activity due to response to ABA, JA, NaCL and PEG treatments.

Author Response

Dear Reviewer;

Thank you so much for your constructive comments on our manuscript The manuscript have been revised carefully.

>Reviewer 3

>The manuscript has improved a lot after the corrections! Adding the supplementary figure 2, which demonstrates that the enzyme activity does not change with the age of the plants in leaves and roots, helps a lot, because it supports the conclusions made about changes in enzyme activity due to response to ABA, JA, NaCL and PEG treatments.

>Thanks you again for your constructive advice and guidance. 

This manuscript is a resubmission of an earlier submission. The following is a list of the peer review reports and author responses from that submission.

Round 1

Reviewer 1 Report

The MS Identification, Characterization, and Stress Responsiveness of Glucose-6-phosphate-Dehydrogenase Genes in Highland Barley brings novel interesting information that is needed. As I know until now, the total number of G6PDH gens in Hordeum vulgare L. has not been published. In addition, authors found out which genes responded to abiotic stressors. I think that this MS is suitable for publication in Plants journal. I have only minor points:

  1. Abstract: It is necessary to explain the abbreviation for NADPH?
  2. Introduction: Citations of involvement of G6PDH in stress: please indicate that there are not presented all published citations or add others.
  3. Introduction: day/night regulation of G6PDH is not mentioned
  4. Methods: Please indicate: the number of biological repetitions and number of plants in each abiotic stress experiment
  5. Methods: What is the principle of the separation of G6PDH activity to cytosolic and plastidic isoform using DTT? It is reduction of disulfide bonds of transit peptide? Does it coincide with the separation by centrifugation in a sucrose gradient?
  6. Fig. 4 Does the abbreviation in the unit of enzyme activity Pro correspond to the protein? Then I recommend title specific activity [nmol NADPH.min-1.mg-1]
  7. Results: p. 7 l. 232, 236 and p. 11, l. 313, 314, 315 If the -fold is lower than 1, it is not increase..e.g. They increased by 0.4-fold.. What does it mean? From 1 to 0.4, it is not increase..
  8. Discussion: The site of action of the inhibitor should be specified (in the watering treatment, or whether it was added only during the measurement)

These comments do not diminish the quality of the work that I recommend for acceptance

Reviewer 2 Report

The main aim of this manuscript was to identify and characterize the stress responses of Glucose-6-phosphate Dehydrogenase Genes (G6PDH), key genes in plant metabolism from Highland Barley. Five G6PDH were cloned and expressed in E. coli to show they are functional. Through GFP fusion and transient expression in tobacco leaves the subcellular localization of these genes was shown. Enzyme activity and transcript analysis under stress showed salt, drought, JA and ABA responses of HvG6PDH1 to HvG6PDH4 in root and leaves. Inhibition of both cytosolic and plastidic G6PDH activities resulted in enhanced accumulation of ROS especially H2O2 and O2- under stress suggesting the role of G6PDH in ROS homeostasis. HvG6PDH2 has been shown to be the unique enzyme in oxidative stress tolerance. The approaches and the experimental designs used by the authors are interesting but except in a different plant species they only confirm the general knowledge on the role of G6PDH under abiotic stresses (Long et al., 2015, Frontiers in plant science7, p.215; Zhao et al., 2020, Frontiers in Plant Science11, p.214). Authors themselves previously identified a G6PDH from Highland Barley (Zhao et al., 2015, J. Plant. Physiol. 2015). In fact, the role of G6PDH2 in stress tolerance has been unequivocally proven through overexpression studies in soybean. No doubt studying G6PDH genes in different species is useful but the findings can be presented in a different way from what has already been known and the new findings of this paper. For example, describing the results/G6PDH’s compared to G6PDH’s from other species in terms of localization, stress response, known function, etc based on the homology would help.

Other comments;

  1. Page 3, line 10: How cDNA was dissolved in DEPC water and adjusted to 50-100nM concentration?
  2. Page 3, line 10: Please check DEPC water concentration (1% or 0.1%).
  3. Manuscript needs professional English editing. For example:

Page 2, line 66: Change “responsing” to “response”.

Page 2, line 89: Remove “literately”.

Page 3, line 128: Change “leaves” to “leaf”.

Page 3, line 132: Remove “Subcellular localization”.

Except in different species, this article is very similar to the following articles

  1. Long, X., He, B., Fang, Y. and Tang, C., 2016. Identification and characterization of the glucose-6-phosphate dehydrogenase gene family in the para rubber tree, Hevea brasiliensis. Frontiers in plant science7, p.215.
  2. Zhao, Y., Cui, Y., Huang, S., Yu, J., Wang, X., Xin, D., Li, X., Liu, Y., Dai, Y., Qi, Z. and Chen, Q., 2020. Genome-Wide Analysis of the Glucose-6-Phosphate Dehydrogenase Family in Soybean and Functional Identification of GmG6PDH2 Involvement in Salt Stress. Frontiers in Plant Science11, p.214.

Reviewer 3 Report

The manuscript represents a well-written and detailed the functional characterization of the enzyme family G6PDH. At some places, a bit more description of how the experiments were conducted and which controls were used, is necessary.

I have the following major and minor comments:  

Major issues:

How was it determined that the isoforms were exactly 5 in highland barley? It should be described.

Which controls were used in the enzymatic activity assay? It should be explained in materials and methods.

The empty vector also shows activity in Figure S1. It should be explained why.

In the legends of the figures, it should be added which statistical test was used. In the materials and methods, only one-way ANOVA is mentioned, but in Figure 8, pairwise tests seem to be done.

What about untreated or mock control in figures 4, 5, 6 and 7? For both cytosolic and plastidic enzymes and for all treatments?

What is the 0 in terms of time of the day in the same figures? Are there any diurnal fluctuations?

Minor issues:

Line 17: I would say “functional analysis” instead of “expression”

Line 64: I would use “protects” or “regulates the response to…” instead of “resists”

Lines 84-85: more explanation is necessary on how the concentrations of NaCl and PEG6000 were determined

Line 85: At what interval were these timepoints (3-120h)?

Line 91: Specify the timepoints

Line 111: At what time of the day were the tissue samples harvested for RNA extraction? Are diurnal changes considered?

Line 113-114: Something wrong with the sentence: or add “that” after “after”, or remove “and”

Line 131: use “were visualized” instead of “was visulized”

Line 145: “were incubated” instead of “was incubated”

Line 153: I would use “measuring” instead of “using”. There is also an unnecessary “was” in the sentence.

Line 167: A verb “is” is missing.

Line 170: remove “the”

Legend of Table 1: Wolf PSORT: add a reference or url;  separate “acidsnumber”; add the name of the online software, which was used for the prediction, not only the url

Line 179: use “were found” instead of “was found”

Line 180: “sequences were found”, not “was found”

Legend Supplementary Figure 1: use “were transformed” instead of “ware”

Line 187: “remarkable” compared to what? Is there any positive control?

Line 208: Avoid using brackets next to each other. The same for lines: 287 and 388

Figure 5D: actin is written in lower case letters and in the other panels, it is in capitals

Line 266: What is the control? Not treated plants or actin?

Figure 6: Which comparisons do the * indicate? It should be described in the legend.

Line 339: “confirmed” what. It is not clear.

Line 346: “Localized in” instead of “localized to”

Line 365: the expression “the key regulator” is too strong. It indeed reduces ROS, but how do these results prove that it is the “key regulator”. There are many redox enzymes, which regulate ROS concentrations in response to stress.

Line 371: I would use “significantly” (if the statistics also confirms it) instead of “obviously”

Line 399: What does it mean “location of harvested material”. Differences in the geographical location, environment, time of harvesting, harvested tissues? Please clarify.

Round 2

Reviewer 2 Report

No attempt was made to describe the findings as suggested. And also, except for the suggested changes in English manuscript has not been completely read and improved. 

Author Response

No attempt was made to describe the findings as suggested. And also, except for the
suggested changes in English manuscript has not been completely read and
improved.

> Thanks for your constructive comments. According to your advice, we revised
carefully the description of results and discussion, and highlighted all the revisions in
violet fronts.

Reviewer 3 Report

The authors have out an effort to improve the manuscript and in the present version the experiments are described much more clearly and the English language has also improved a lot.

I still have one major consideration about the controls. As I understand from the description in materials and methods, the plants are grown until 5 day old and then they are treated with NaCL, PEG, ABA and JA for 3, 6, 12, 24, 48, 72, 96, 120, 144h. At each of these timepoints, plant material from the treated plants is harvested to determine G6PDHs' expression and activity in different tissues. The expression and activity values for each timepoint are always compared to the untreated plant at timepoint 0. My problem with this experimental design is that the age of the plant is not taken into account and this could also possibly affect the gene expression and enzyme activity of G6PDH. For example the plants treated with ABA for 72h are already 8 days old and should be compared to untreated or mock-treated plants, which are also 8 days old and not to the starting point (5day-old untreated plant).  In my view, it is more correct to have untreated or mock-treated plants, which grow in parallel with the treated ones and are harvested at the same time-points as the treated ones. Like this, one will have a control for each of the time-points, which would allow to eliminate all other factors influencing the difference in the gene expression and enzyme activiy, thus all differences would be due to the treatment.

Round 3

Reviewer 2 Report

In this version authors made an attempt to improve English. The manuscript can be accepted with minor English corrections such as Fig. 1 legend and Fig. 3: "Hvv"